# Diagnostic limitations of clinical case definitions of pertussis in infants and children with severe lower respiratory tract infection

**Rudzani Muloiwa** [1]*, **Mark P. Nicol**[2,3], **Gregory D. Hussey**[4,5], **Heather J. Zar**[6,7]

**1** Department of Paediatrics & Child Health, Groote Schuur Hospital, University of Cape Town, Cape Town, South Africa, **2** Division of Medical Microbiology, Faculty of Health Sciences, University of Cape Town, Cape Town, South Africa, **3** Division of Infection and Immunity, School of Biomedical Sciences, University of Western Australia, Perth, Australia, **4** Institute of Infectious Disease & Molecular Medicine, University of Cape Town, Cape Town, South Africa, **5** Division of Medical Microbiology, Vaccines for Africa Initiative, University of Cape Town, Cape Town, South Africa, **6** SA-MRC unit on Child & Adolescent Lung Health, University of Cape Town, Cape Town, South Africa, **7** Department of Paediatrics & Child Health, Red Cross War Memorial Children's Hospital, University of Cape Town, Cape Town, South Africa

* Rudzani.Muloiwa@uct.ac.za

## Abstract

### Introduction

Diagnosis of pertussis is challenging especially in infants. Most low and middle-income countries (LMIC) lack resources for laboratory confirmation, relying largely on clinical diagnosis alone for both case management and surveillance. This necessitates robust clinical case definitions.

### Objectives

This study assesses the accuracy of clinical case definitions with and without lymphocytosis in diagnosing pertussis in children with severe lower respiratory tract infection (LRTI) in a LMIC setting.

### Methods

Children hospitalized with severe LRTI in a South African hospital were prospectively enrolled and evaluated for pertussis using PCR on respiratory samples. Clinical signs and differential white cell counts were recorded. Sensitivity and specificity of pertussis clinical diagnosis using WHO and Global Pertussis Initiative (GPI) criteria; and with addition of lymphocytosis were assessed with PCR as the reference standard.

### Results

458 children <10 years were enrolled. *Bordetella pertussis* infection was confirmed in 32 (7.0%). For WHO criteria, sensitivity was 78.1% (95% CI 60.7–89.2%) and specificity 15.5% (95% CI 12.4–19.3%); for GPI sensitivity was 34.4% (95% CI 20.1–52.1) and specificity 64.8% (95% CI 60.1–69.2%). Area under the curve (AUC) on receiver operating character (ROC) analysis was 0.58 (95% CI 0.46–0.70 for WHO criteria, and 0.72 (95% CI 0.56–0.88)

**Data Availability Statement:** All relevant data are within the manuscript and its Supporting Information files.

**Funding:** The authors acknowledge financial support to their institution by Sanofi Pasteur for the submitted work. The funders had no role in study design, data collection and analysis, decision to publish, or preparation of the manuscript.

**Competing interests:** The authors acknowledge financial support to their institution by Sanofi Pasteur for the submitted work. This does not alter our adherence to PLOS ONE policies on sharing data and materials. Rudzani Muloiwa has received honoraria from both Sanofi Pasteur and Pfizer for speaking engagements.

for GPI with highest likelihood ratios of 5.33 and 4.42 respectively. Diagnostic accuracy was highest between five and seven days of symptoms for both criteria. Lymphocytosis had sensitivity of 31.3% (95% CI 17.5–49.3%) and specificity of 70.7% (95% CI 66.1–74.8%) and showed a marginal impact on improving clinical criteria.

## Conclusion

Clinical criteria lack accuracy for diagnosis and surveillance of pertussis. Non-outbreak settings should consider shorter durations in clinical criteria. New recommendations still fall short of what is required for a viable clinical screening test which means the need to improve access to laboratory diagnostic support remains crucial.

## Introduction

Pertussis has resurged globally over the last decade. Waning immunity change in primary schedule to acellular vaccines and improvement in diagnostics have been given as possible explanations.[1, 2] A large proportion of reported cases have not shown the classic presentation of a prolonged spasmodic cough with inspiratory whoop and post tussive vomiting.[3] In response, some high income countries (HICs) have modified their case definitions to suit local diagnostic and surveillance criteria; usually this includes reducing duration of cough symptoms to lower than the two weeks historically recommended as the minimum cut-off time.[4]

The World Health Organization (WHO) developed criteria for clinically defining cases of pertussis. These include presence of a cough for at least 14 days characterized by one of paroxysms, inspiratory whoop or post-tussive vomiting, Table 1.[5] Apart from reduction by a week in the duration of cough from more than 21 days, very little has changed over the last three decades in the WHO case definition of pertussis.[6]

**Table 1. Clinical features for diagnosis of pertussis cases.**

| World Health Organization | Global Pertussis Initiative | |
|---|---|---|
| ■ A case diagnosed as pertussis by a physician, <u>OR</u> | ■ Cough or illness in a person without or with only minimal fever <u>AND:</u> | |
| ■ A person with a cough lasting ≥2 weeks with ≥1 of the following: symptoms: | **0–3 months** | **4 months to 9 years** |
|  | Cough and coryza **PLUS** any of | Paroxysmal cough **PLUS** any of |
| • Paroxysms (i.e., fits) of coughing | • Whoop | • Whoop |
| • Inspiratory "whooping" | • Apnea | • Apnea |
| • Post-tussive vomiting (i.e., vomiting | • Post-tussive emesis | • Post-tussive emesis |
| immediately after coughing) without | • Cyanosis | • Worsening of symptoms at night |
| other apparent cause | • Seizure | • Seizure |
|  | • Pneumonia | • Pneumonia |
|  | • Close exposure to an adolescent or adult (usually a family member) with a prolonged afebrile cough illness | • Close exposure to an adolescent or adult (usually a family member) with a prolonged afebrile cough illness |

Adapted from World Health Organization & Cherry et al (2012)[4, 5] **NB.** Global Pertussis Initiative criteria for individuals older than 9 years not shown

Acknowledging shortcomings of WHO clinical case definition, the Global Pertussis Initiative (GPI), following a Roundtable discussion in 2010, recommended changes to improve diagnostic sensitivity and specificity for pertussis. In addition to clinical features found in WHO criteria, the GPI suggests adding presence of coryza, apnea, seizures or cyanosis as well as absence of fever, all without specified duration. GPI clinical features vary by age categories: less than four months of age, four months to nine years and 10 years and above, Table 1.[4]

Although culture and ELISA can be used to confirm pertussis, polymerase chain reaction (PCR) has gained favor as the most practical confirmatory method with an acceptable level of sensitivity and specificity.[7, 8] In contrast to HICs which have access to laboratory resources to validate clinical suspicion, most low and middle-income countries (LMICs) lack resources for laboratory confirmation, relying largely on clinical diagnosis alone.[9] It is therefore imperative to have a robust clinical definition in these settings.

Pertussis is commonly associated with leukocytosis, with an increase in lymphocytes.[10] In the absence of resources for confirming pertussis, clinicians have also used lymphocytosis to support a clinical diagnosis.[11]

Here we investigate the sensitivity and specificity, in a non-outbreak setting, of clinical criteria recommended by WHO and GPI to diagnose pertussis with PCR as the diagnostic reference standard, in a LMIC. The analysis includes assessment of the impact of duration of symptoms on sensitivity and specificity for both sets of diagnostic criteria. Secondarily, we assess the impact of adding lymphocytosis to clinical criteria to improve diagnostic accuracy.

## Materials and methods

The study was approved by the Human Research Ethics Committee of the Faculty of Health Sciences of the University of Cape Town; Reference: 371/2011. Written informed consent was sought and received from the legal guardian for the participation of each into the study.

A prospective study was conducted from September 2012 to September 2013, to investigate the incidence of pertussis in children hospitalized for severe lower respiratory tract infection (LRTI), or with apnea, at a tertiary hospital in South Africa. Children were included if they presented with WHO defined severe LRTI (cough or difficulty breathing and age specific tachypnoea plus chest indrawing and/or presence of danger signs necessitating hospital admission) [12] or apnea after written informed consent was received from the parent or legal guardian. Participants were excluded if they had been admitted in a health care facility in the preceding two weeks, or if a legal guardian was not available to sign consent. A nasopharyngeal swab and an induced sputum specimen were taken from each child and sent to the laboratory for culture and pertussis PCR.

The duration and character of cough, especially presence of paroxysms, inspiratory whoop, or post-tussive vomiting; as well as presence of apnea, cyanosis, seizures and fever were recorded. In addition, any prior exposure to antibiotics including macrolides and cotrimoxazole were noted.

A blood cell count was done including white cell count with a differential. An absolute count of more than 9000 cells/μL or 7000 cells/μL was used to define lymphocytosis in infancy or in children 12 months and older respectively, according to local laboratory guidelines.

### Pertussis PCR

PCR targeting the IS481 common Bordetella insertion site and IS1001 for *Bordetella parapertussis* was done. Specimens testing positive for IS481 were further tested with PCR for hIS1001 to exclude *Bordetella holmesii* which shares the same IS481 insertion site. IS481+/ hIS1001- samples were classified as confirmed *Bordetella pertussis* infection. The methods and epidemiological findings of this study have been published elsewhere.[13] Only assessment of *Bordetella pertussis* was included in the current analysis to allow for comparison with other studies.

### Analysis of data

Data were analyzed using STATA statistical package version 14 (StataCorp, College Street, Texas).

Proportions were summarized as percentages with a 95% interval of confidence, as indicated. Age in months and duration of symptoms in days were summarized using medians and interquartile ranges.

The Wilcoxon rank sum test was used to test the difference between continuous variables while the χ2 or Fisher's exact tests were used to compare proportions of categorical variables as appropriate. Where hypothesis testing was undertaken the P-value was set at a two-tailed $p<0.05$ as a cut-off point of statistical significance.

Proportions of cases conforming to positive and negative clinical case definitions were compared with PCR, the reference standard, to respectively estimate sensitivity and specificity of using clinical signs in the diagnosis of pertussis. Clinical features, chosen to conform to WHO and GPI recommendations (Table 1), were assessed with and without taking duration of symptoms into consideration in this analysis.

As our sample did not include older individuals, the analysis was restricted to children less than 10 years of age who presented with respiratory illness and underwent confirmatory diagnostic testing for pertussis.

Although GPI criteria do not include duration, for comparison, both WHO and GPI criteria were additionally assessed for sensitivity and specificity at different durations of symptoms. A receiver operating character (ROC) analysis was done to determine area under the curve (AUC) with PCR as diagnostic reference standard. Finally, the utility of adding lymphocytosis to WHO and GPI criteria was analyzed.

Our data allowed us to stratify analysis to reflect only two of the GPI suggested age groups: below four months of age and four months to nine years of age.

## Results

Four hundred and fifty-eight children were enrolled with a median age of 8 (IQR 4–17) months; 132 (28.8%) were younger than 4 months of age. HIV infection was confirmed in 19 (4.2%) while 92 (20.1%) were exposed to HIV in utero but tested negative for HIV infection. Forty-five children (9.8%) were classified as moderately to severely malnourished using WHO criteria. Table 2.

A total of 32 (7.0%, 95% CI 4.8–9.7%) cases were confirmed to have *Bordetella pertussis* on PCR, including 13 (9.9%) of the 132 infants younger than four months and 19 (5.8%) of 326 in the older age group. One of the PCR confirmed cases was also positive on culture. Lymphocytosis was found in 135 (29.5%) of which the two age groups had 33/132 (25.0%) and 102/326 (31.3%) respectively. Clinical features of PCR positive and negative cases were similar across age groups, Table 2. Only 3 (9%) of the 32 PCR positive participants were suspected to have pertussis by the attending clinicians while 12 (2%) of the PCR negative participants were diagnosed with pertussis; P = 0.079.

The median duration of symptoms was 3 (IQR 2–5) days. Paroxysmal cough was reported as a presenting feature in 398 (69.9%) children; similar in confirmed pertussis cases and those with negative PCR under four months of age with 8/13 (61.5%) and 79/119 (66.4%) respectively; p = 0.726. Similarly, paroxysmal cough was found in 15/19 (79.0%) participants four months to nine years old with confirmed pertussis compared to 218/326 (71.0%) with negative PCR; P = 0.837. The other presenting clinical features are shown in Table 3.

Of the clinical features recommended by WHO for diagnosing pertussis, presence of paroxysms had the highest sensitivity (61.5% 0 to 3 months and 78.9% 4 months to 9 years,

**Table 2. Baseline characteristics of study participants (N = 458).**

|  | n (%) |
|---|---|
| **PCR confirmed cases** | 32 (7.0) |
| **Lymphocytosis** | 135 (29.5) |
| **Age group** | |
| 0–3 months | 132 (28.8) |
| 4 months– 9 years | 372 (71.2) |
| **Sex** | |
| Female | 200 (43.7) |
| **Pertussis vaccine doses** | |
| 0 | 28 (6.1) |
| 1 | 57 (12.4) |
| 2 | 58 (12.6) |
| $\geq 3$ | 308 (67.2) |
| Unknown | 7 (1.5) |
| **HIV status** | |
| Infected | 19 (4.1) |
| Exposed uninfected | 92 (20.1) |
| **Nutritional status** | |
| Moderate-severe malnutrition# | 45 (9.8) |
| **Macrolide/cotrimoxazole in preceding week** | 5 (1.1) |

PCR = polymerase chain reaction, # As per World Health Organization criteria of weight for age Z score less than -2

respectively) while inspiratory whoop gave the highest specificity (68.9% % 0 to 3 months and 76.2% 4 months to 9 years, respectively) of any single feature on its own, Fig 1. Overall, when the whole group was considered, the use of any feature suggested by WHO had sensitivity of

**Table 3. Clinical presentation of children by *Bordetella pertussis* PCR status N = 458.**

|  | Clinical Feature | PCR+[n (%)] | PCR- [n (%)] | P Value |
|---|---|---|---|---|
|  |  | **n = 13** | **n = 119** |  |
|  | Paroxysmal cough | 8(61.5) | 79(66.4) | 0.726 |
|  | Whoop | 4(30.8) | 37(31.1) | 1.000 |
|  | Apnea | 2(15.4) | 7(5.9) | 0.217 |
|  | Post-tussive emesis | 7(53.9) | 56(47.1) | 0.642 |
| **0–3 months** | Cyanosis | 2(15.4) | 3(2.5) | 0.076 |
|  | Seizure | 0(0.0) | 1(0.8) | 1.000 |
|  | Pneumonia | 5(38.5) | 35(29.4) | 0.532 |
|  | Absence of fever | 3(23.1) | 33(27.7) | 1.000 |
|  |  | **n = 19** | **n = 326** |  |
|  | Paroxysmal cough | 15(79.0) | 218(71.0) | 0.457 |
|  | Whoop | 4(21.1) | 73(23.8) | 1.000 |
|  | Apnea | 1(5.3) | 10(3.3) | 0.489 |
| **4 months—9 years** | Post-tussive emesis | 5(26.3) | 130(43.4) | 0.169 |
|  | Seizure | 2(10.5) | 6(2.0) | 0.073 |
|  | Night cough | 15(79.0) | 239(78.0) | 1.000 |
|  | Pneumonia | 6(31.6) | 109(35.5) | 0.728 |
|  | Absence of fever | 5(26.3) | 142(46.3) | 0.090 |

PCR = polymerase chain reaction

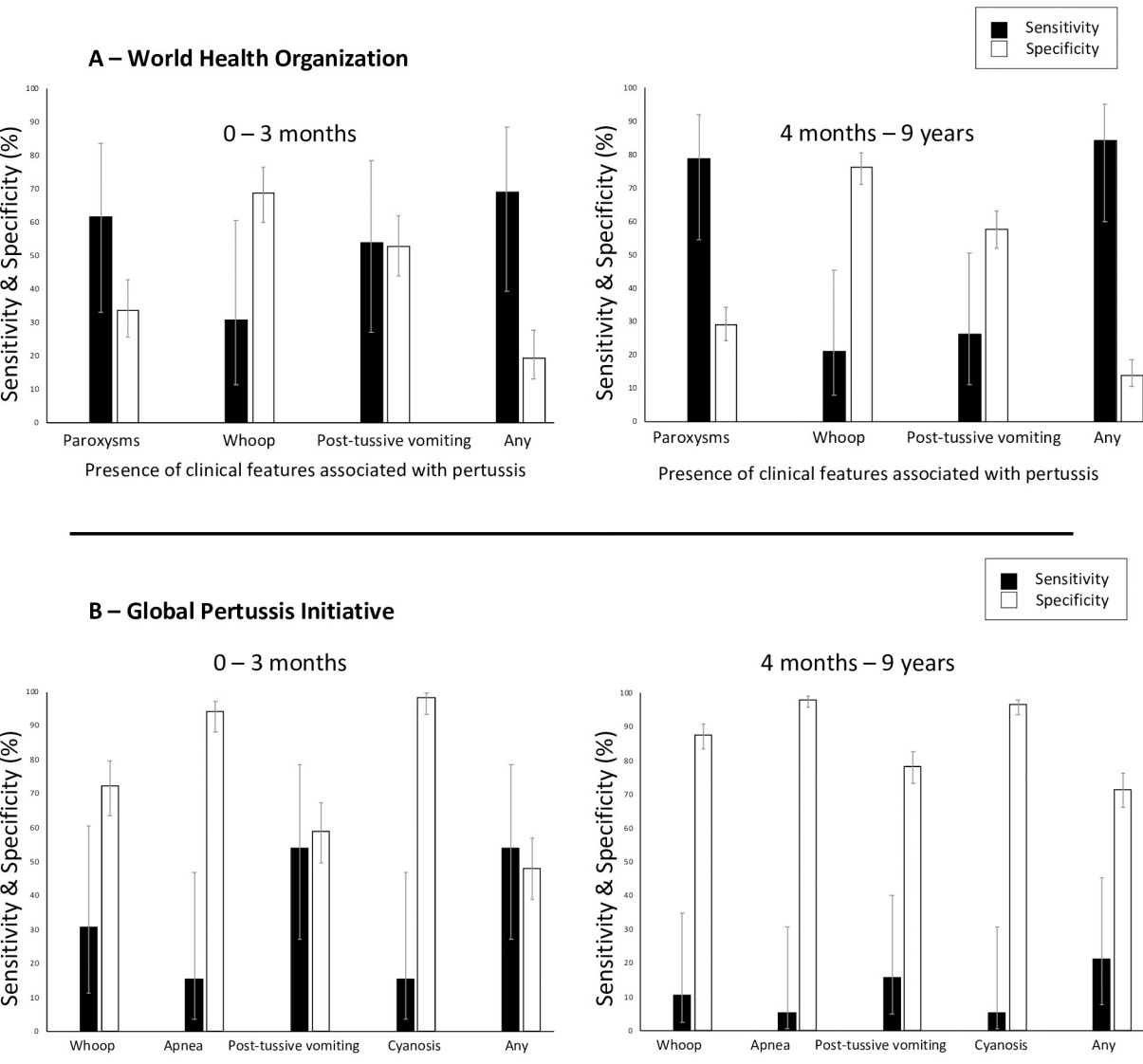

**Fig 1. Sensitivity and specificity of clinical features in the diagnosis of pertussis.**

78.1% (95% CI 60.7–89.2%) while GPI suggested clinical features had sensitivity of 34.4% (95% CI 20.1–52.1%). Specificity was 15.5% (95% CI 12.4–19.3%) and 64.8% (95% CI 60.1–69.2%) for WHO and GPI, respectively. Fig 1.

When ROC analysis was undertaken on duration of symptoms, GPI recommended features had AUC of 0.72 (95% CI 0.56–0.88) while those suggested by WHO had AUC of 0.58 (95% CI 0.46–0.70). The AUC was greater in younger infants than in the older age group for both WHO and GPI criteria. Fig 2.

Specificity increased for both WHO and GPI criteria as duration of symptoms increased while the opposite was seen with sensitivity declining with duration. The decline in sensitivity occurred earlier and steeper when using WHO clinical criteria than when GPI criteria were utilized. Sensitivity was 12.0% and 27.3% at ≥14-day duration for WHO and GPI criteria respectively, while specificity was 94.0% and 90.7% respectively for WHO and GPI criteria at a similar duration cut-off.

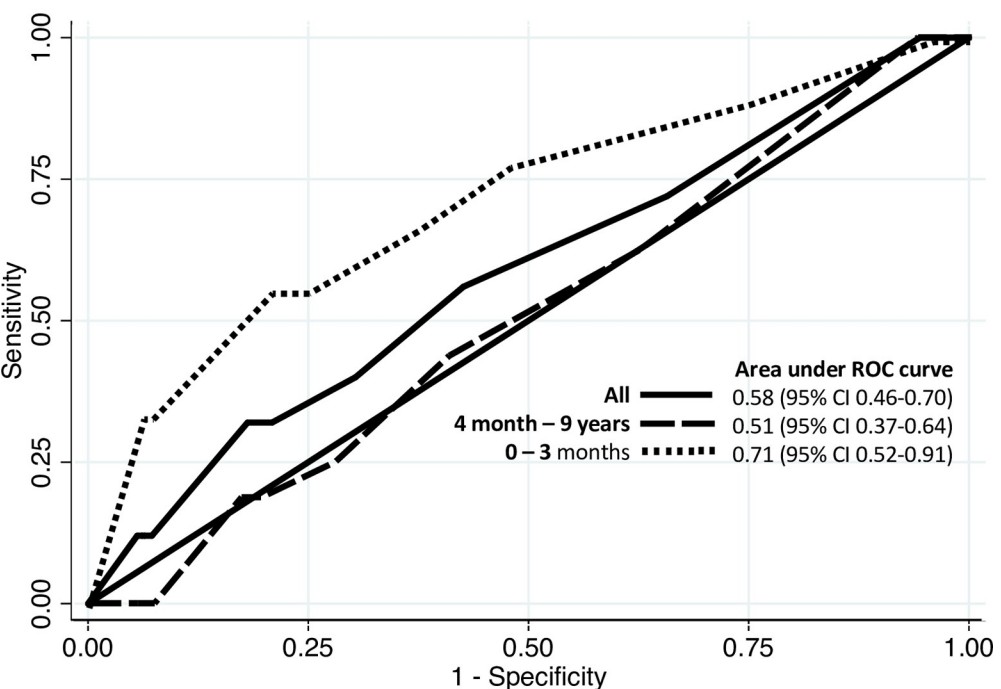

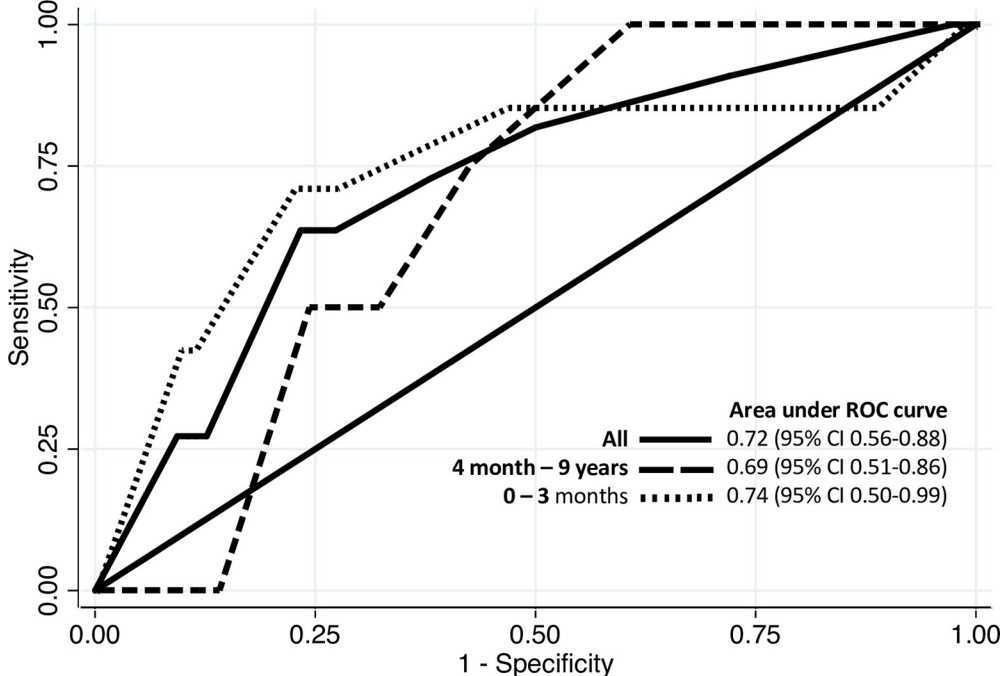

**Fig 2. Receiver Operating Characteristics (ROC) curves for duration of symptoms.**

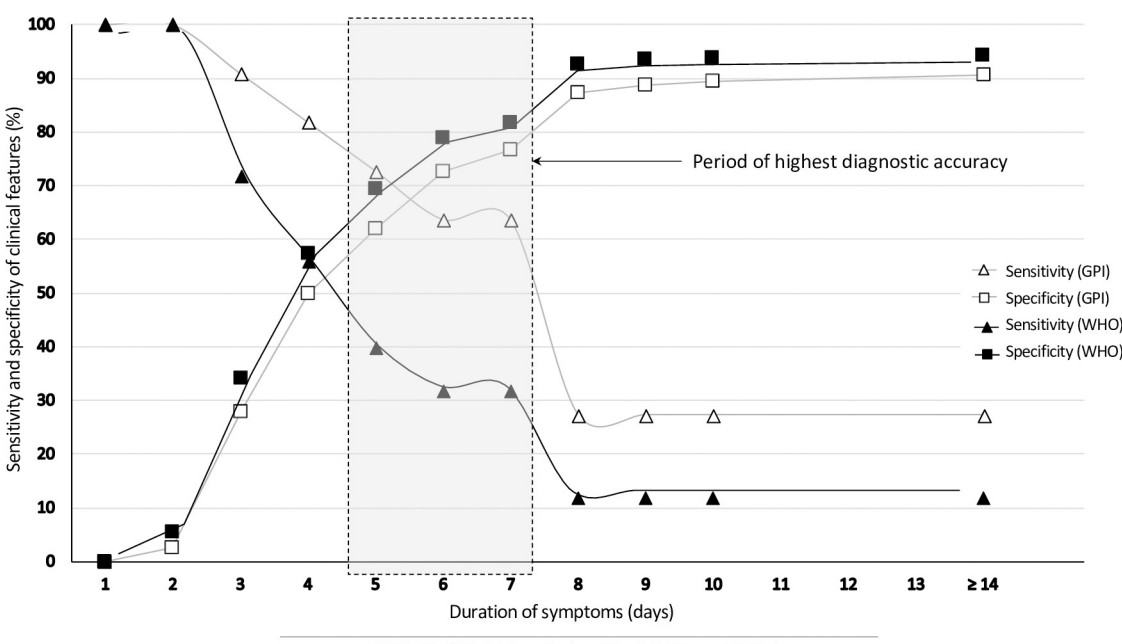

**Fig 3. Sensitivity and specificity of clinical features in the diagnosis of pertussis with changing duration**

In general, sensitivity declined, and specificity increased with increase in duration of symptoms. When any symptom was considered, the highest combination of sensitivity and specificity with the use of GPI clinical features was seen between five and seven days with sensitivity ranging between 63.6% and 72.7% and specificity ranging between 62.0% and 76.7%. Criteria recommended by WHO showed their highest combination of sensitivity and specificity between three and five days duration of symptoms with sensitivity ranging between 40.0% and 72.0%, and specificity ranging between 34.3% and 69.6%. Fig 3. The highest likelihood ratio for sensitivity was seen in the young infant group. This ranged between 1.00 and 5.33 using WHO based criteria and between 0.96 and 4.42 when GPI based criteria were applied.

Lymphocytosis when used alone had a sensitivity of 31.3% (95% CI 17.5–49.3%) and a specificity of 70.7% (95% CI .66.1–74.8%) compared to PCR. The sensitivity and specificity when lymphocytosis was combined with WHO clinical criteria were 36.0% (95% CI 19.6–56.4%) and 75.6% (95% CI 63.7–84.8%) for the whole group, respectively; while combining GPI criteria with lymphocytosis gave sensitivity and specificity of 45.5% (95% CI 20.1–73.4%) and 71.4% (95% CI 65.7–76.4%), respectively. Age stratified sensitivity and specificity of lymphocytosis when combined with WHO and GPI criteria is shown in Fig 4.

## Discussion

Pertussis remains difficult to diagnose clinically with certainty as demonstrated by low sensitivity and specificity using current clinical case definitions. Adding lymphocytosis had a marginal impact in improving the diagnosis of pertussis in cases preselected with the use of WHO and GPI clinical criteria. In addition, the low sensitivity was across age groups.

The overall sensitivity and specificity of clinical features were generally low irrespective of whether WHO or GPI diagnostic criteria were used. Criteria recommended by WHO showed better sensitivity than those suggested by the GPI when duration of symptoms was not considered, but the latter showed better specificity. Paroxysmal cough and post-tussive vomiting as

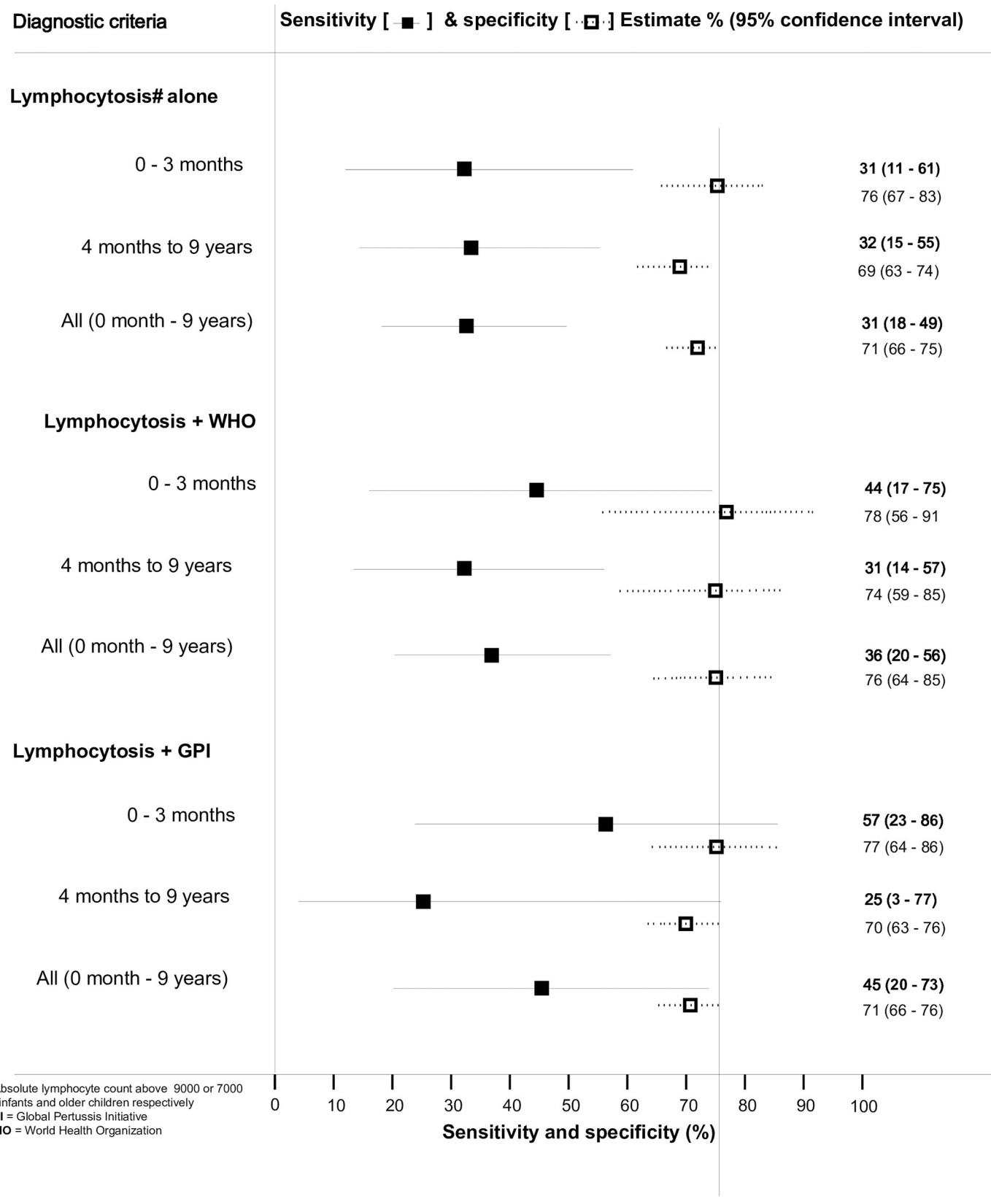

**Fig 4. Sensitivity and specificity of lymphocytosis in the diagnosis of pertussis.**

standalone clinical features, gave the best sensitivity for both sets of criteria. The highest likelihood ratios of just above five and four for WHO and GPI based criteria, respectively, are too low to make for a functional diagnostic tool for clinical practice.

A systematic review conducted on the use of symptoms in the diagnosis of pertussis concluded that the presence of whooping or post-tussive vomiting could be used to make a possible diagnosis of pertussis in adults while the absence of paroxysmal cough would possibly exclude it.[14] This study concluded that post-tussive vomiting was less helpful as a clinical diagnostic test in children. However, when Cornia *et al* analyzed the likelihood ratios of clinical features used in the same systematic review, they found all positive likelihood ratios to be less than 2, significantly lower than a threshold regarded robust enough for clinical use.[15] A similar likelihood was observed when data from other systematic reviews on sensitivity of clinical signs in the diagnosis of pertussis were analyzed.[16, 17] Our study with its best likelihood ratios of four and five for WHO and GPI criteria, respectively, still fell short of ratios of at least 10, that would be required for a useful clinical diagnostic tool.[15]

In our study, when duration of cough was added to the other clinical features, GPI criteria had overall better diagnostic accuracy compared to WHO as indicated by the higher AUC (0.72 versus 0.58). For both criteria, duration based diagnostic accuracy was better in young infants than in older infants and children (AUC 0.71 vs. 0.51 and AUC 0.74 vs. 0.68 for WHO and GPI, respectively). Increase in the duration of symptoms reduced the sensitivity of clinical diagnosis in both sets of criteria, while the opposite was noted with respect to specificity. The best combinations of sensitivity and specificity (60% to 70% for both sets of criteria) were seen between five- and seven-days duration of symptoms. This suggests that suspected cases of pertussis, in children hospitalized with LRTI or apnea, are more likely to be correctly classified as positive or negative in children with this symptom duration. However, early diagnosis is desirable for effective treatment and to prevent transmission

An Iranian study reported 95% sensitivity and 15% specificity using a 14 day cut-off with at least one WHO clinical criterion in a cohort of children and adolescents 6 to 14 years of age. [18] Duration of symptoms equal to or longer than 14 days gave a specificity of 63% and sensitivity of 93% in outbreak settings in one American study.[19] A recently published Serbian study testing diagnostic of GPI criteria in individuals older than 3 months reported sensitivity ranging between 5% and 76% in the 4 months to 9 year old group and between 2% and 73% in the older group; while for both groups specificity ranged between 50% and 100%.[20] For each clinical feature evaluated, there was an inverse relationship between sensitivity and specificity; a pattern we observed throughout the current study. Unlike in our study, all the studies mentioned here seem to have selected participants on the bases of clinical criteria some of which were later tested for their sensitivity, which may explain the high sensitivity reported in some of them.

Adding lymphocytosis to our analysis marginally improved the diagnostic sensitivity of suspected cases screened with either criteria but showed a specificity higher than 70% for both. The increase in sensitivity was higher in younger infants than in older children for both WHO and GPI when lymphocytosis was added to the criteria. This suggests that although lymphocytosis may be less useful in confirming the diagnosis of pertussis, when used age stratification should be considered. Very high lymphocyte counts have been shown to predict the severity and likelihood of dying in infants with pertussis; and therefore, should best be utilized for this role in the management of pertussis rather than diagnosis.[10, 21]

The difficulty involved in the diagnosis of pertussis has major implications for clinical management, infection control, surveillance and the conduct of pertussis vaccine trials. Missed diagnoses of pertussis due to low sensitivity affects both appropriate clinical intervention in the index patient and contacts as well as appropriate reporting and surveillance. In addition,

low sensitivity undermines optimizing of preventive measures including immunization. On the other hand, over diagnosis of pertussis may lead to overtreatment and inappropriate use of resources. Overall, diagnostic shortcomings have serious economic implications, particularly in poor resourced settings. Clinical and surveillance practice favors higher sensitivity at the expense of specificity, but low specificity has a huge impact on the estimated efficacy of pertussis vaccines.[6] There may be a need to develop context specific definitions of pertussis rather than attempting to come with one for all settings.

Our study was limited by a small sample size as well as the use of a single laboratory confirmation method performed at a single point in the course of the illness. As the diagnostic sensitivity of PCR is not constant throughout the course of illness, a negative result at a single point does not necessarily exclude infection with *B. pertussis*.[9, 22] We were also unable to assess the impact of HIV on clinical criteria as the sample size was not sufficient to make such an analysis. In addition, as our study population consisted of children with severe disease requiring hospitalization, these findings may not be generalizable to children with less severe disease.

While acknowledging the observed poor diagnostic accuracy of clinical criteria, the slight improvement observed with use of GPI criteria should encourage development of screening criteria. Screening with a high sensitivity (but acceptably low specificity) triage algorithm before laboratory testing would potentially reduce the waste of testing patients that are likely to return a negative result, while increasing the sensitivity of confirmatory PCR. Finding such screening criteria will require well-designed, prospective studies specifically investigating clinical diagnosis of pertussis. Such studies should ideally employ different laboratory diagnostic methods with testing at multiple periods in the course of the illness and include appropriate controls to allow estimation of specificity. A majority of published studies exploring sensitivity and specificity of pertussis clinical diagnosis are done in participants included on clinical suspicion of pertussis. Involving only suspected cases of pertussis to assess diagnostic accuracy of the same features on which the sample was selected has the potential of exaggerating the diagnostic accuracy of clinical criteria.[23]

## Conclusions

Ultimately, in the absence of laboratory confirmation, confident diagnosis of pertussis on clinical grounds is difficult even for experienced physicians, especially when cases do not have the classical presentation. In our study, only 9% of confirmed cases were suspected to have pertussis, substantially underestimating pertussis as a possible cause of severe respiratory illness in our cohort. There is no obvious substitute for pertussis laboratory confirmation, and more effort is needed to increase this resource in LMIC settings where most pertussis is speculated to occur.[3, 24] In the current presence of limited resources, it would seem prudent to consider using modified case definitions, such as the one suggested by the GPI.[25] It is clear that longer durations of symptoms in a number of criteria used all over the world—including the ones suggested by WHO and the Centre for Disease control—greatly undermine sensitivity.[5, 26] Due consideration must be given to abandon these in favour of shorter durations of five to seven days used in conjunction with age-stratified criteria, even if this is at the expense of over-diagnosing pertussis.

## Supporting information

**S1 Data.**
(PDF)

## Acknowledgments

We would like to acknowledge the participants, laboratory personnel and study staff for their immense contribution.

## Author Contributions

**Conceptualization:** Rudzani Muloiwa, Gregory D. Hussey, Heather J. Zar.

**Data curation:** Rudzani Muloiwa, Mark P. Nicol.

**Formal analysis:** Rudzani Muloiwa.

**Funding acquisition:** Gregory D. Hussey.

**Methodology:** Rudzani Muloiwa.

**Supervision:** Gregory D. Hussey, Heather J. Zar.

**Writing – original draft:** Rudzani Muloiwa.

**Writing – review & editing:** Rudzani Muloiwa, Mark P. Nicol, Gregory D. Hussey, Heather J. Zar.

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
