## [Decision Letter · Decision Letter 0]

24 Apr 2020

PONE-D-20-03797

Diagnostic limitations of clinical case definitions of pertussis in infants and children with severe lower respiratory tract infection

PLOS ONE

Dear Dr Rudzani Muloiwa,

Thank you for submitting your manuscript to PLOS ONE. After careful consideration, we feel that it has merit but does not fully meet PLOS ONE’s publication criteria as it currently stands. Therefore, we invite you to submit a revised version of the manuscript that addresses the points raised during the review process.

We would appreciate receiving your revised manuscript by May 2. To enhance the reproducibility of your results, we recommend that if applicable you deposit your laboratory protocols in protocols.io, where a protocol can be assigned its own identifier (DOI) such that it can be cited independently in the future. For instructions see: http://journals.plos.org/plosone/s/submission-guidelines#loc-laboratory-protocols

We look forward to receiving your revised manuscript.

Kind regards,

Daniela Flavia Hozbor

Academic Editor

PLOS ONE

Journal Requirements:

"The authors acknowledge financial support to their institution by Sanofi Pasteur for the submitted work.  The funders had no role in study design, data collection and analysis, decision to publish, or preparation of the manuscript."  

We note that you received funding from a commercial source: 'Sanofi Pasteur'

Reviewers' comments:

Reviewer's Responses to Questions

**Comments to the Author**

1. Is the manuscript technically sound, and do the data support the conclusions?

Reviewer #1: Yes

Reviewer #2: Yes

2. Has the statistical analysis been performed appropriately and rigorously? 

Reviewer #1: Yes

Reviewer #2: Yes

3. Have the authors made all data underlying the findings in their manuscript fully available?

Reviewer #1: Yes

Reviewer #2: Yes

4. Is the manuscript presented in an intelligible fashion and written in standard English?

Reviewer #1: Yes

Reviewer #2: Yes

5. Review Comments to the Author

Reviewer #1: The manuscript addresses a topic that has been widely covered but with no final conclusions. I am referring to the difficulties encountered with the clinical diagnosis of pertussis. Even without providing a solution to these difficulties, it brings an interesting analysis of them. The manuscript is correctly written, it is easy to read, methods are clearly described and the results are easily visualized. The conclusions are correctly based on the results and I especially highlight the good analysis of the limitations of the paper.

Reviewer #2: The main aim of the manuscript is to propose an improvement to the GPI (Global Pertussis Initiative) experts pertussis case definition to increase the sensitivity and specificity and though enhancing pertussis surveillance and diagnosis in low and middle-income countries which lack resources for laboratory confirmation of pertussis cases. Clinical case definitions with higher sensibilities are specially useful and important to change for countries other than developed ones. Comparison with WHO clinical case definition is also included.

This prospective study analyzes clinical symptoms along with pertussis specific PCR confirmatory results in two groups of children under 9 years of age (0 - 3 months and 4 months to 9 years) diagnosed with severe lower tract respiratory infection or apnea.

Since GPI clinical case definition does not include for these age groups a duration of coughing, the authors demonstrate a the better accuracy in pertussis diagnosis using GPI clinical case definition that incorporates a duration from 5 to 7 days of cough.

To my opinion the authors conclusions and proposal should strictly be mentioned to be used for children admitted to hospital facilities with severe low respiratory tract infections or apnea. This should be reflected along the conclusion section e.g. modify lines 268-72. Although clarified in lines 313-15 it should be highlighted before.

It would be nice to include WHO definition for severe lower tract infections for readers who are not clinicians or pediatricians (does it includes pneumonia and bronchitis?).

Specially in the context of classical pertussis clinical presentation I wonder the impact of the proposal in regions where pertussis cases present with upper respiratory tract infection. Is this a situation possibly being observed in low and middle-income countries?

Line 81: direct fluorescent antigen testing is no longer recommended for laboratory confirmation

Line 101: was the time of patients enrollment an outbreak year? please clarify

- Was there an exclusion criteria for patients enrollment used? please clarify

Lines 118-21: english should be improved.

Line 123: reference 12 refers to the use of cotton swabs for Bordetella pertussis culture. Since B. pertussis growth is inhibited by long chain fatty acids present in cotton, it may be possible that positivity by culture is low because of this reason. What kind of swabs do you use?

Lines 252-60: I suggest to add a connection sentence with the idea behind this evidence supporting the importance of your results.

Line 262: duration is referred to cough duration please clarify

Lines 274-84: What is the idea behind incorporating references 17,18 and 19? Is it to include other age groups with higher specificities and sensibilities? is that used for comparison purposes? 282-84: why do authors suggest that? please clarify

6. PLOS authors have the option to publish the peer review history of their article (what does this mean?). If published, this will include your full peer review and any attached files.

Reviewer #1: No

Reviewer #2: No

---

## [Author Response · Author response to Decision Letter 0]

3 Jun 2020

Reviewer's comments

NB. Due to a well-known Microsoft Word glitch, some line numbers are ‘lost’, skipped at the start of new pages in the tracked document. This has led to different numbering between the tracked and untracked documents. The line numbers referring to the changes made are those of the untracked document.

Reviewer #1: 

Reviewer #1 comment 1:

The manuscript addresses a topic that has been widely covered but with no final conclusions. I am referring to the difficulties encountered with the clinical diagnosis of pertussis. Even without providing a solution to these difficulties, it brings an interesting analysis of them. The manuscript is correctly written, it is easy to read, methods are clearly described, and the results are easily visualized. The conclusions are correctly based on the results and I especially highlight the good analysis of the limitations of the paper.

Response to reviewer #1 comment 1

Thank you for your encouraging comment. We really appreciate it.

Reviewer #2: 

Reviewer #2 comment 1:

The main aim of the manuscript is to propose an improvement to the GPI (Global Pertussis Initiative) experts pertussis case definition to increase the sensitivity and specificity and though enhancing pertussis surveillance and diagnosis in low and middle-income countries which lack resources for laboratory confirmation of pertussis cases. Clinical case definitions with higher sensibilities are especially useful and important to change for countries other than developed ones. Comparison with WHO clinical case definition is also included.

This prospective study analyzes clinical symptoms along with pertussis specific PCR confirmatory results in two groups of children under 9 years of age (0 - 3 months and 4 months to 9 years) diagnosed with severe lower tract respiratory infection or apnea.

Since GPI clinical case definition does not include for these age groups a duration of coughing, the authors demonstrate a the better accuracy in pertussis diagnosis using GPI clinical case definition that incorporates a duration from 5 to 7 days of cough.

To my opinion the authors conclusions and proposal should strictly be mentioned to be used for children admitted to hospital facilities with severe low respiratory tract infections or apnea. This should be reflected along the conclusion section e.g. modify lines 268-72. Although clarified in lines 313-15 it should be highlighted before.

Response to reviewer #2 comment 1:

Thank you for this comment. We appreciate it and are greatly encouraged. To address the specific issue of highlighting the current population, an amendment has been made on lines xx to xx. The section now reads, “This suggests that suspected cases of pertussis, in children hospitalized with LRTI or apnea, are more likely to be correctly classified as positive or negative in children with this symptom duration.” Line 273

Reviewer #2 comment 2:

It would be nice to include WHO definition for severe lower tract infections for readers who are not clinicians or pediatricians (does it includes pneumonia and bronchitis?).

Response to reviewer #2 comment 2:

We have included the WHO definition we used and added a reference. The modified section in Material and Methods now reads, “Children were included if they presented with WHO defined severe LRTI (cough or difficulty breathing and age specific tachypnoea plus chest indrawing and/or presence of any danger sign necessitating hospital admission) or apnea after written informed consent was received from the parent or legal guardian” [12] Lines 103 – 104

A reference [Lines 399 – 400] has also been added:

12. Handbook: integrated management of childhood illnesses. Geneva: World Health Organization; 2005. 

Please note that these criteria for classifying severity have since undergone further modification.

Reviewer #2 comment 3:

Specially in the context of classical pertussis clinical presentation I wonder the impact of the proposal in regions where pertussis cases present with upper respiratory tract infection. Is this a situation possibly being observed in low and middle-income countries?

Response to reviewer #2 comment 3:

Thank you for the comment. Mild cases have indeed been observed, for example as seen in surveillance data from acute influenza-like illnesses. For this study we wanted to see how diagnostic criteria hold in the case of severe cases – the sort of cases that lead to more resources being required and in which deaths are a potential outcome. We have noted this as a limitation of our study by stating, “In addition, as our study population consisted of children with severe disease requiring hospitalization, these findings may not be generalizable to children with less severe disease.” Lines 317 – 319

Reviewer #2 comment 4:

Line 81: direct fluorescent antigen testing is no longer recommended for laboratory confirmation

Response to reviewer #2 comment 4:

Thank you for pointing this out. The reference to direct fluorescent antigen has been removed. The sentence now reads, “Although culture and ELISA can be used to confirm pertussis, polymerase chain reaction (PCR) has gained favor as the most practical confirmatory method with an acceptable level of sensitivity and specificity.[7, 8]” Lines 81 – 83 

Reviewer #2 comment 5:

Line 101: was the time of patients enrollment an outbreak year? please clarify

Response to reviewer #2 comment 5:

The year was not an outbreak year. We have added a section to clarify this. Line xx now reads, 

“Here we investigate the sensitivity and specificity, in a non-outbreak setting, of clinical criteria recommended by WHO and GPI to diagnose pertussis with PCR as the diagnostic reference standard, in a LMIC.” Lines 92 – 94

Reviewer #2 comment 6:

Was there an exclusion criteria for patients enrollment used? please clarify

Response to reviewer #2 comment 6:

Thank you for pointing this out. We have amended the section to make explicit our exclusion criteria by stating, “Participants were excluded if they had been admitted in a health care facility in the preceding two weeks, or if a legal guardian was not available to sign consent.” Lines 106 – 107 

Reviewer #2 comment 7:

Lines 118-21: English should be improved.

Response to reviewer #2 comment 7:

After rereading the section, we can see the ambiguity created by our use of language. The sentence has been broken into two to improve clarity. The section now reads, “PCR targeting the IS481 common Bordetella insertion site and IS1001 for Bordetella parapertussis was done. Specimens testing positive for IS481were further tested with PCR for hIS1001 to exclude Bordetella holmesii which shares the same IS481 insertion site.” Lines 121 – 123

Reviewer #2 comment 8:

Line 123: reference 12 refers to the use of cotton swabs for Bordetella pertussis culture. Since B. pertussis growth is inhibited by long chain fatty acids present in cotton, it may be possible that positivity by culture is low because of this reason. What kind of swabs do you use?

Response to reviewer #2 comment 8:

We use rayon and Dacron swabs in our setting, this study used the latter. Although we agree that the type of swab could impact on culture yield, we think that the quality of the swabs would not have increased the detection rate to exceed that of PCR, hence our adoption of PCR as the reference standard.

Reviewer #2 comment 9:

Lines 252-60: I suggest to add a connection sentence with the idea behind this evidence supporting the importance of your results.

Response to reviewer #2 comment 9:

Thank you for this advice. In order to relate our study to the previous section, we have added the following sentence, “Our study with its best likelihood ratios of four and five for WHO and GPI criteria, respectively, still fell short of ratios of at least 10, that would be required for a useful clinical diagnostic tool.[15]” Lines 261 – 263

The reference at the end of the sentence has already been used previously in the text:

15. Cornia PB, Lipsky BA. Symptoms Associated With Pertussis Are Insufficient to Rule In or Rule Out the Diagnosis. Chest. 2019;155(2):449-50. Epub 2019/02/09. doi: 10.1016/j.chest.2018.10.028. PubMed PMID: 30732694. Lines 409 – 411

Reviewer #2 comment 10:

Line 262: duration is referred to cough duration please clarify

Response to reviewer #2 comment 10:

Thank you for pointing this omission out. We have clarified this by adding the word ‘cough’. The sentence now reads, “In our study, when duration of cough was added to the other clinical features, …” Line 265

Reviewer #2 comment 11:

What is the idea behind incorporating references 17,18 and 19? Is it to include other age groups with higher specificities and sensibilities? is that used for comparison purposes? 282-84: why do authors suggest that? please clarify

Response to reviewer #2 comment 11:

The three references were included to demonstrate the difficulty consistently encountered of having to make very ‘expensive’ trade-offs between sensitivity and specificity, a pattern that we also observed in our study. The following line has been added to clarify this, “For each clinical feature evaluated, there was an inverse relationship between sensitivity and specificity; a pattern we observed throughout the current study.” Lines 284 – 285

With the lines highlighted above (282-284) we wanted to show that when cases are preselected to undergo a laboratory confirmatory test using clinical features that are in keeping with pertussis when the same laboratory test is used also as gold standard to test ‘sensitivity’ of the same features, it should not be surprising to have an observation of higher sensitivity than in studies such as ours, in which cases are not similarly preselected on clinical suspicion of pertussis. We have amended the sentence to clarify this, “Unlike in our study, the three studies mentioned here seem to have selected participants on the bases of pertussis clinical criteria some of which were later tested for their sensitivity, which may explain the high sensitivity reported in some of them.” Lines 285 – 288

---

## [Editor Report · Decision Letter 1]

22 Jun 2020

Diagnostic limitations of clinical case definitions of pertussis in infants and children with severe lower respiratory tract infection

PONE-D-20-03797R1

Dear Dr. Rudzani Muloiwa,

We’re pleased to inform you that your manuscript has been judged scientifically suitable for publication and will be formally accepted for publication once it meets all outstanding technical requirements.

Kind regards,

Daniela Flavia Hozbor

Academic Editor

PLOS ONE
---

## [Editor Report · Acceptance letter]

6 Jul 2020

PONE-D-20-03797R1 

Diagnostic limitations of clinical case definitions of pertussis in infants and children with severe lower respiratory tract infection 

Dear Dr. Muloiwa:

I'm pleased to inform you that your manuscript has been deemed suitable for publication in PLOS ONE. Congratulations! Your manuscript is now with our production department. 

Kind regards, 

on behalf of

Dr. Daniela Flavia Hozbor 

Academic Editor

PLOS ONE